# SEER: A Knapsack approach to Exemplar Selection for In-Context HybridQA

**Jonathan Tonglet**[1,2], **Manon Reusens**[2], **Philipp Borchert**[2,3], **Bart Baesens**[2,4]

[1]Ubiquitous Knowledge Processing Lab (UKP Lab),
Department of Computer Science and Hessian Center for AI (hessian.AI),
TU Darmstadt
[2] Research Centre for Information Systems Engineering (LIRIS),
Faculty of Economics and Business, KU Leuven
[3]IESEG School of Management, 3 Rue de la Digue, 59000 Lille, France
[4]Department of Decision Analytics and Risk, University of Southampton
www.ukp.tu-darmstadt.de

## Abstract

Question answering over hybrid contexts is a complex task, which requires the combination of information extracted from unstructured texts and structured tables in various ways. Recently, In-Context Learning demonstrated significant performance advances for reasoning tasks. In this paradigm, a large language model performs predictions based on a small set of supporting exemplars. The performance of In-Context Learning depends heavily on the selection procedure of the supporting exemplars, particularly in the case of HybridQA, where considering the diversity of reasoning chains and the large size of the hybrid contexts becomes crucial. In this work, we present **S**election of **ExE**mplars for hybrid **R**easoning (**SEER**), a novel method for selecting a set of exemplars that is both representative and diverse. The key novelty of SEER is that it formulates exemplar selection as a Knapsack Integer Linear Program. The Knapsack framework provides the flexibility to incorporate diversity constraints that prioritize exemplars with desirable attributes, and capacity constraints that ensure that the prompt size respects the provided capacity budgets. The effectiveness of SEER is demonstrated on FinQA and TAT-QA, two real-world benchmarks for HybridQA, where it outperforms previous exemplar selection methods[1].

## 1 Introduction

Hybrid documents, which combine tables and text paragraphs, are prevalent in various industries such as finance, healthcare, and manufacturing. The development of question-answering systems capable of effectively handling these documents holds the potential to automate business processes and enhance accessibility to the information they contain. Several benchmarks anchored in the financial domain have been introduced for hybrid question answering (HybridQA) (Chen et al., 2021c; Zhu et al., 2021; Zhao et al., 2022). Figure 1 presents an example from the FinQA dataset. Despite ongoing progress, current HybridQA models have yet to achieve human expert performance (Zhu et al., 2021; Chen et al., 2021c).

Recently, In-Context Learning (ICL) with Large Language Models (LLMs) has shown great performance on reasoning tasks. In this setting, a set of exemplars, i.e. training instances with their answers, are provided as part of the input prompt to assist LLMs in generating the correct answer. ICL is an inference time technique that keeps the LLMs parameters frozen (Brown et al., 2020). The performance of ICL depends heavily on the quality of the provided exemplar set (Liu et al., 2022a). Various strategies, ranging from random selection to similarity-based retrieval, have been proposed to tackle this problem (Liu et al., 2022b; Rubin et al., 2022; Li and Qiu, 2023; Lu et al., 2023). When selecting exemplars for HybridQA, special consideration must be given to the unique challenges of the task, including diverse reasoning chains, large context sizes (Chen et al., 2021c; Zhao et al., 2022), and the limited correlation between the question and its reasoning chain.

In this work, we propose Knapsack Programs as a framework to model exemplar selection for ICL. Knapsacks are a family of Integer Linear Programs that search an optimal subset of items under linear constraints (Wolsey, 2020). For a given test instance, a Knapsack Program is solved to obtain the optimal exemplar set. This expressive framework

---

[1]Code available at github.com/jtonglet/SEER

| | 2013 | 2014 | 2015 | 2016 | 2017 |
|---|---|---|---|---|---|
| Cme group inc. | $ 164.01 | $ 194.06 | $ 208.95 | $ 279.85 | $ 370.32 |
| S&P 500 | 132.39 | 150.51 | 152.59 | 170.84 | **208.14** |
| Peer group | 176.61 | 187.48 | 219.99 | 249.31 | 323.23 |

[…] an investment of $ **100** ( with reinvestment of all dividends ) is assumed to have been made in our class a common stock , […] s&p 500 peer group $ **100** invested on 12/31/12 in stock or index , […]

**Question :** what is the anualized return for s&p 500 from 2012 to 2017?

**Answer :**
**Subtract(208.14, 100); Divide(#0, 100);**
**Divide(1,5); Exp(#1,#2); Subtract(#3,1)**

**Python code :**
```
table_query_0 = 208.14
text_variable_0 = 100

step_0 = table_query_0 – text_variable_0
step_1 = step_0 / table_query_0
step_2 = 1 / 5
step_3 = step_1**step_2
ans = step_3 - 1
```

Figure 1: An instance of the FinQA dataset (Chen et al., 2021c). Text snippets of the text paragraphs are shown in the dashed box.

allows balancing the diversity and similarity of the selected exemplars while controlling the prompt size with user-defined linear constraints. We introduce SEER, a novel method to select exemplars for HybridQA using Knapsack Programs. SEER reduces the candidate set with a nearest neighbor filtering, and leverages constraint modules to predict the attributes of the test instance. The attributes of a HybridQA instance are properties that influence the underlying reasoning chain, e.g., the modality (table, text, hybrid) and the answer type (span extraction, arithmetic reasoning, counting). By leveraging constraint modules, SEER shapes the Knapsack structure to prioritize the selection of exemplars that share similar attributes with the test instance.

The contributions of this work are as follows: (1) we introduce Knapsack Programs as a framework for ICL exemplar selection. (2) We propose SEER, a novel exemplar selection method for In-Context HybridQA. (3) We address all three challenges of HybridQA exemplar selection at the same time with fine-grained token budget constraints and attribute-guided selection. Extensive evaluation of two real-world HybridQA benchmarks shows that SEER outperforms state-of-the-art exemplar selection methods, especially under restricted token capacity budgets.

## 2 Related work

**In-Context Learning** LLMs have shown the ability to perform a wide range of tasks with only a few exemplars provided as a prompt while keeping all the model parameters frozen (Brown et al., 2020). However, the performance is highly dependent on the quality of the provided exemplars

(Zhao et al., 2021; Lu et al., 2022). Hence, several approaches have been explored for exemplar selection, including nearest neighbor search (Liu et al., 2022a), reinforcement learning (Zhang et al., 2022; Lu et al., 2023), clustering (Zhang et al., 2023), search algorithms (Li and Qiu, 2023), and supervised learning (Rubin et al., 2022; Ye et al., 2023). Rubin et al. (2022) consider token capacity indirectly by constructing the largest possible prompt with the selected exemplars. In contrast to previous methods, we consider exemplar selection as an ILP Knapsack Program, which allows us to specify diversity-enhancing constraints, directly optimize the token capacity without simple heuristics, and leverage powerful solvers to find a performant exemplar selection.

**Hybrid Question Answering** Chen et al. (2020, 2021b) introduced the task of HybridQA on open-domain Wikipedia pages. Later on, datasets based on real-world financial documents were introduced (Zhu et al., 2021; Chen et al., 2021c; Zhao et al., 2022; Chen et al., 2022b). Previous work has focused on improving the retriever-generator framework (Lei et al., 2022; Sun et al., 2022; Zhang and Moshfeghi, 2022). Chen et al. (2022a) use few-shot chain-of-thoughts to solve the task of HybridQA. However, they focus on improving the prompt format, while our work focuses on selecting good exemplars.

**Integer Linear Programming for NLP** Integer Linear Programming has been used in many NLP tasks (Martins, 2014), including coreference resolution (Denis and Baldridge, 2007; De Belder and Moens, 2012), sentence compression (De Belder and Moens, 2010), dependency parsing (Riedel and Clarke, 2006), semantic role labeling (Roth and

Yih, 2005), and translation (Germann et al., 2004). In this work, we introduce a novel application of ILP in NLP: exemplar selection for ICL. Furthermore, it is, to the best of our knowledge, the first time that the Knapsack family of ILP programs is used in NLP.

## 3 Methodology

### 3.1 Integer Linear Programming

Linear Programming (LP) involves maximizing or minimizing an objective function while adhering to a set of constraints. The objective function consists of a weighted linear combination of variables, while the constraints are (in)equalities that involve linear combinations of these variables. These constraints serve to restrict the value ranges of the variables and capture their interaction effects (De Belder and Moens, 2010). Integer Linear Programming (ILP) is a subset of LP, wherein variables are constrained to take only integer values. ILP is divided into program families, one of which is the Knapsack. Given a set of items, the Knapsack's objective is to select the subset that maximizes the total value while remaining within the maximum capacity (Wolsey, 2020).

More formally, the problem can be expressed as:

$$\text{maximize} \quad \sum_{i \in S} w_i \ x_i$$

$$\text{subject to} \quad \sum_{i \in S} c_i \ x_i \leq C \quad i \in S$$
$$x_i \in \{0, 1\} \qquad i \in S$$

where $x_i$ is a variable that takes value 1 when item $i$ in set $S$ is selected, 0 otherwise. $w_i$ and $c_i$ are parameters representing the value and cost of item $i$, respectively. $C$ is the maximum capacity of the Knapsack. Depending on the setting, additional constraints and variables can be added to the basic Knapsack program template shown above. The Knapsack is NP-hard, but several algorithms have been developed to find an optimal solution efficiently, including Branch-and-Bound and Cutting Planes (De Belder and Moens, 2010). Several solvers provide efficient implementations of these algorithms (Martins, 2014).

### 3.2 Challenges of Exemplar Selection for HybridQA

When solving HybridQA problems with ICL, the task is to predict an answer $A$ given a question

| Question | Program | Answer | Modality | Answer type |
|---|---|---|---|---|
| What is the **percentage change in cost of hardware** between 2019 and 2018? | / | ['-10%'] | table | span |
| What was the **percentage change in cost of software** under development between 2018 and 2019? | (16,284 − 6,509) / 6,509 | 150.18 | table | arithmetic |
| On an average how many people are **employed** in R&D in fiscal in 2019? | / | ['9000'] | text | span |
| What is the average number of Administrative Staff **employed**? | (798+784+833) / 3 | 805 | table | arithmetic |

Figure 2: Four problem instances from TAT-QA. Similar questions do not always share similar problem attributes.

$Q$, a hybrid context consisting of text paragraphs $P$ and a table $T$, and a set of $n$ exemplars $E = \{e_1, ..., e_n\}$ where $e_i$ is a tuple $(Q_i, A_i, P_i, T_i)$.

$$A = argmax_a P(a \mid Q, P, T, E)$$

Prior studies (Liu et al., 2022a; Lu et al., 2023) have demonstrated that the thoughtful selection of the exemplar set $E$ can improve and stabilize the performance of ICL over a random selection baseline. However, selecting the optimal exemplar set poses three challenges for HybridQA problems. First, there is a high diversity in the type of questions and in the approaches to solve them. The financial dataset FinQA for example contains more than 300 different numerical formulas. For questions asking to compute a "percentage change", the training set counts 12 unique formulas. Given this diversity, it is not possible to cover all possible reasoning chains with a single set of exemplars. This challenge is partially addressed by similarity-based exemplar selection methods (Liu et al., 2022a).

Secondly, Figure 2 shows the additional challenge of the low correlation between the problem's question and attributes. This might result in prediction errors, as these problems seem semantically similar, however, they require different modalities and answer types.

Thirdly, HybridQA problems, especially when dealing with real-world data like financial documents, involve large contexts. LLMs have a limit to the number of input and output text tokens they can process, whether it is due to organizational resource constraints or the inherent limitation of the LLM. Consequently, it becomes crucial to ensure that tokenized exemplars fit within the LLM's capacity while reserving enough tokens for generating the

desired output text. In the following, we propose a new exemplar selection method that addresses those three challenges by modelling them as objectives and constraints of a Knapsack program. Notably, this work addresses explicitly the latter two challenges for the first time, contributing to the advancement of exemplar selection in this domain.

## 3.3 SEER

SEER generates Knapsack programs for exemplar selection in HybridQA. Given the training set and a test instance, SEER constructs a unique Knapsack program using nearest neighbor filtering and constraint modules. The selected exemplar set is the optimal solution to the Knapsack program. These exemplars, along with the test instance, are provided as prompts to an LLM for predicting the final answer. Figure 3 provides an overview of SEER's methodology.

**Similarity computation** involves calculating cosine similarity between pairs of HybridQA problems' questions in the embedding space. The resulting similarity values serve as coefficients in the objective function. To ensure accurate comparisons, preprocessing is applied to remove noise. (1) All questions are lowercased. (2) All punctuation is removed. (3) Dates, numerical values, locations, and companies are replaced by their NER tags.

**Nearest Neighbor filtering** involves applying an initial filter (Liu et al., 2022a) to identify the $k$ candidates from the training set that exhibit substantial surface similarity with the test instance, thus narrowing down the search space.

**Constraint modules** predict an attribute of the test instance. We define an attribute as a characteristic of the reasoning chain of a HybridQA problem. Attributes include modality, answer type, and number of reasoning steps. Inferring these attributes involves a standard classification task, where the question and hybrid context are provided as input, and the output corresponds to one of the attribute values. The task can be addressed through fine-tuning or ICL.

A SEER Knapsack is uniquely defined for a test instance by the combination of the similarity weights and the predicted attributes. As an illustration, the base template for Knapsack programs with predicted attribute "modality:table" is formulated as follows, where variable $x_i$ takes value 1 if instance $i$ from the candidate set $S$ is selected as an exemplar, 0 otherwise. The candidate set $S$ is a subset of

the original training set, composed of the $k$ nearest neighbor of the test instance's. The Knapsack has a double capacity constraint. Firstly, $M$ is the maximum allowed number of exemplars. Secondly, $L$ is the maximum combined length of the exemplars $l_i$ in number of tokens. The value of $L$ depends on the backbone LLM.

$$
\begin{aligned}
\text{maximize} \quad & \sum_{i \in S} w_i \; x_i \\
\text{subject to} \quad & \sum_{i \in S} x_i \leq M \\
& \sum_{i \in S} l_i \; x_i \leq L \\
& \sum_{i \in S} table_i \; x_i \geq \alpha \, M \\
& \sum_{i \in S} other_i \; x_i \geq \beta \, M \\
& x_i \in \{0, 1\} \qquad i \in S \\
\text{where} \quad & other_i = text_i + hybrid_i
\end{aligned}
$$

The program contains two or more diversity constraints. The structure of those constraints depends on the attributes predicted by the constraint modules. All possible diversity constraint configurations are presented in Appendix B. Parameter $\alpha$ ranges from 0 to 1 and controls the promotion of exemplars possessing the predicted attribute, i.e. "table" in our case. Parameter $\beta$, also in the range [0,1], controls the promotion of the other attributes. It holds that $\alpha + \beta \leq 1$. $tab_i$, $text_i$, and $hybrid_i$ are binary parameters with value 1 when the modality of candidate $i$ is the table, the text, or both, respectively, and 0 otherwise. The objective is to maximize the sum of the exemplar's similarity with the test instance $w_i$, while respecting all constraints. Framing exemplar selection as a Knapsack program has several benefits. It is solvable with efficient deterministic algorithms. It gives full control to the end user over the prompt capacity in tokens and explicitly allows the user to balance the similarity and diversity of the selected exemplars in terms of the test instance's attributes. SEER's pseudo-code is shown in Appendix A.

## 3.4 LLM Code Generation

The ICL exemplars selected by SEER and the test instance are concatenated and provided as a prompt to an LLM. To remove irrelevant text paragraphs from the hybrid context in relation to the question, we employ a pre-trained text retriever. Recent

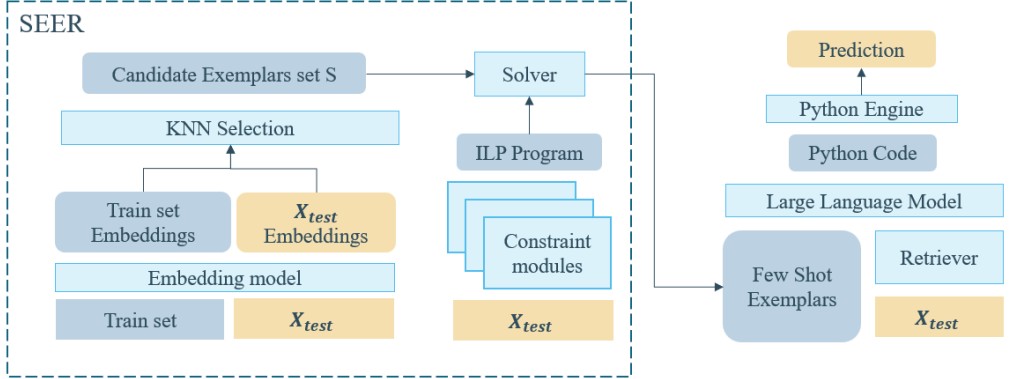

Figure 3: Overview of SEER's architecture to select the optimal exemplar set for a HybridQA problem.

studies have demonstrated the benefits of formulating answer derivation as Python code (Chen et al., 2022a; Gao et al., 2023; Mishra et al., 2022). Inspired by these approaches, we adopt a code generation formulation instead of text generation. The resulting code is executed using an external engine to derive the answer. Figure 1 depicts the conversion from the original text answer to a Python code answer.

## 4 Experiments

### 4.1 Datasets

We evaluate SEER on two HybridQA datasets with real-world contexts: FinQA (Chen et al., 2021c) and TAT-QA (Zhu et al., 2021), both anchored in the financial domain.

**FinQA** comprises 8,281 problems, each containing a context with a table and multiple text paragraphs. The answers in FinQA are expressed in a domain-specific language as a sequence of operators with two operands each. These expressions are then translated to Python code using an automated script, with each operation on a separate line. Tables are linearized with " | " as column delimiter and " \n " as row delimiters. A prompt example is provided in Figure 1. This dataset includes a constraint module for predicting the modality.

**TAT-QA** consists of 16,552 problems. Since the ground truth answers for the test set of TAT-QA are not publicly available, we employ the original dev set as the test set. To compensate, we create a new dev set of equal size by splitting the training set. The context consists of text paragraphs and a table. Some tables are flattened, while others exhibit complex structures with multiple header levels. The original answers are written as text or equations. We convert text to Python comments.

Equations are written as a one-line variable assignment. A prompt example is provided in Appendix C. Unlike FinQA which only contains arithmetic problems, TAT-QA includes other answer types such as (multi-)span extraction and counting problems. Consequently, TAT-QA has two constraint modules, one for modality and one for the answer type.

The evaluation of the datasets is based on their respective metrics: execution accuracy (EA) and program accuracy (PA) (Chen et al., 2021c) for FinQA, and Exact Match (EM) and numeracy-focused F1 (Dua et al., 2019) for TAT-QA. EA and EM assess the correctness of the final answer, while PA ensures that the generated code is mathematically equivalent to the ground truth derivation. The numeracy-focused F1 (Dua et al., 2019) computes the F1 score over the bag-of-word representation of the generated answer. For TAT-QA, we exclude the sub-task of scale prediction and reserve it for future work in multi-task ICL HybridQA. The LLM has little exposure to syntactic conventions unrelated to correct reasoning, such as determining the appropriate scale to represent a percentage ([0,1] or [0,100]). To take this into account, we allow some flexibility in the answer evaluation script. For further details on the evaluation procedure, refer to Appendix E.

### 4.2 Baselines

We compare SEER with other exemplar selection strategies for ICL. The **Random** baseline randomly selects a set of exemplars from the training set for each test instance. We define **CSP** (Constraint Satisfaction Problem) as SEER without the objective function. A candidate set is randomly selected among those that meet all the Knapsack's con-

straints. The **Fixed set** baseline employs the same set of exemplars for every test instance. Specifically, we randomly sample 20 groups of 4 and 8 exemplars from the training set for FinQA and TAT-QA, respectively. The group exhibiting the highest performance on the dev sets is selected. **KATE** (Liu et al., 2022a) selects the k nearest neighbors to the test instance. It corresponds to SEER without token capacity and diversity constraints. **Diverse KATE**, an extension of KATE, divides the training set into two subsets, one for text problems and one for table problems. An equal number of nearest neighbors is selected from each subset. This ensures that both modalities are present in the exemplar set. **PromptPG** (Lu et al., 2023) is a reinforcement learning method that trains a policy network to select well-performing exemplars among a fixed candidate pool. At inference time, the policy decides which exemplars from the candidate set are selected for a given test instance. For training, we use the same parameters as Lu et al. (2023), except the size of the candidate exemplar set which is set to 20 and 40 for FinQA and TAT-QA respectively. To disentangle the exemplar selection performance from errors related to constraint modules, we also include results obtained using ground truth problem attributes, denoted as **SEER**$_{gold}$, which serves as an upper bound estimate for SEER's performance. Although our primary focus is on ICL, numerous studies have explored fine-tuning approaches for HybridQA. We report the results of the SOTA fine-tuning approach. For a comparison of fine-tuning approaches with SEER, refer to Appendix F.

### 4.3 Implementation details

We use CODEX, a 175 Billion parameters LLM available through the OpenAI API[2] (Chen et al., 2021a), as a backbone for SEER and all baselines. CODEX has demonstrated exceptional performance in code generation tasks (Chen et al., 2021a). For the computation of question and text paragraph embeddings, we employ Sentence-BERT models (Reimers and Gurevych, 2019). Regarding the constraint modules, we explore two strategies. Firstly, fine-tuning a BERT model on the training set. Secondly, employing ICL predictions with one exemplar per attribute value, leveraging CODEX as the underlying model. The ICL strategy offers the advantage of requiring only a small number of training instances with attribute labels.

The ICL prompt starts by the text paragraphs, the linearized table and the problem's question, in that order. Then, a task-specific instruction is provided at the end of the prompt. Instruction for **modality prediction**: "Do you need data from the table, the text paragraphs, or both (hybrid) to answer this question? Answer by one of the following: table, text, hybrid.". Instruction for **answer type prediction**: "Does this question require to extract spans from the document, to count, or to perform an arithmetic reasoning? Answer by one of the following: span, multi-span, count, arithmetic.". We set the maximum token length $L$ to the maximum capacity of the LLM minus the token lengths of the problem's context and the longest answer in the training set. This conservative estimate of $L$ ensures that the exemplars, the problem's context, and its answer can all fit within the imposed token limit. We use the GUROBI solver[3] to find the optimal Knapsack solution. The detailed parameter values are listed in Appendix D. Our experiments are run on a single NVIDIA GeForce RTX 3050 GPU, except for CODEX inferences, which are performed on the OpenAI servers.

### 4.4 Main results

Table 1 shows the main results, showcasing the superior performance of SEER compared to all exemplar selection baselines. Notably, SEER$_{gold}$ achieves better results than SEER by a margin of 0.4 to 1.5%, indicating that accurate attribute prediction by the constraint modules significantly contributes to the overall performance. The difference between KATE and SEER on TAT-QA is not significant. However, SEER$_{gold}$ is significantly different from KATE, showing that SEER ultimately outperforms KATE with better constraint modules. For the detailed significance test of the main results, please refer to Appendix G. The lower performance of random selection reaffirms the necessity of careful exemplar selection. Despite its relatively low performance, CSP demonstrates the notable improvements achievable by introducing constraints into the random selection process, surpassing the outcomes of a purely random approach. Interestingly, Diverse KATE is marginally stronger than KATE on FinQA but 7% lower in EM on TAT-QA. While the SOTA fine-tuning models outperform ICL methods, SEER contributes to closing the gap. The average run time distribution is as follows:

[2]https://openai.com/blog/openai-codex

[3]https://www.gurobi.com/solutions/gurobi-optimizer

|  | FinQA | | TAT-QA | |
| **Method** | **EA** | **PA** | **EM** | **F1** |
| --- | --- | --- | --- | --- |
| PromptPG (Lu et al., 2023) | 53.56 ± 3e-3 | 24.09 ± 1e-3 | 51.64 ± 0.27 | 58.86 ± 0.27 |
| Random | 55.65 ± 3.35 | 29.5 ± 17.91 | 49.7 ± 1.38 | 57.31 ± 1.24 |
| CSP | 60.77 ± 0.14 | 43.62 ± 0.23 | 57.47 ± 0.19 | 65.21 ± 0.13 |
| Fixed set | 64.05 ± 0.22 | 38.15 ± 0.08 | 66.55 ± 0.18 | 73.8 ± 0.11 |
| KATE (Liu et al., 2022a) | 67.07 ± 0.04 | 58.65 ± 0.04 | 68.9 ± 0.07 | 75.77 ± 0.08 |
| Diverse KATE | 67.31 ± 0.24 | 59.54 ± 0.19 | 61.53 ± 0.23 | 68.89 ± 0.21 |
| SEER | 68.85 ± 0.04 | 59.78 ± 0.15 | 69.68 ± 0.07 | 76.71 ± 0.07 |
| $\text{SEER}_{gold}$ | 69.25 ± 0.11 | 60.16 ± 0.14 | 71.32 ± 0.07 | 78.12 ± 0.08 |
| SOTA fine-tuned model | 71.07 | 68.94 | 73.6 | 81.3 |

Table 1: Table of main results (%). EA indicates the Execution Accuracy, PA the Program Accuracy, EM the Exact Match, and F1 the numeracy-focused F1 score. Reported results are averages over three iterations on the test set. CODEX is the backbone LLM for ICL methods.

solving the Knapsack with Gurobi takes 0.2 seconds, predicting an attribute requires 0.05 seconds with BERT and 2.95 seconds with CODEX, and predicting the answer with CODEX takes 2.89 seconds, on average.

Figure 4 presents a comparison between SEER and two strong baselines across four subsets of FinQA test instances. The set of instances with attributes correctly predicted by the constraint module, called Correct Attribute Predicted (CAP), and its complement, Incorrect Attribute Predicted (IAP). The set of instances with at least one selected exemplar having the correct attribute, called Correct Attribute Selected (CAS), and its complement, Incorrect Attribute Selected (IAS). It holds that $\text{CAP} \subseteq \text{CAS}$, $\text{IAS} \subseteq \text{IAP}$, and $\text{CAS} \cap \text{IAP} \neq \varnothing$. We conclude that SEER's hedge over baselines is the direct consequence of the prediction and selection of correct attributes. Interestingly, the baselines struggle with instances where incorrect attributes are predicted and selected, indicating a correlation between the difficulty of a HybridQA instance and the challenge of predicting its attributes. Where no correct attribute is selected (IAS), the Fixed set baseline outperforms SEER and KATE by a slight margin.

### 4.5 Constraint modules results

Table 2 shows the performance of fine-tuned and ICL constraint modules on the dev and test sets. On two out of the three attribute prediction tasks, the fine-tuned BERT module performs best. However, the ICL constraint module's performance is always close, making it a viable alternative when attribute labels are not available in the training set. There is still room to improve the performance of

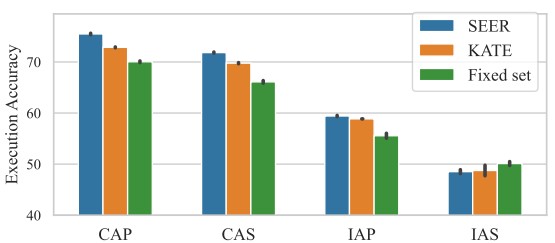

Figure 4: EA (%) on four subsets of the FinQA test instances. Results are averages over three iterations. CAP stands for Correct Attribute Predicted, CAS for Correct Attribute Selected, IAP for Incorrect Attribute Predicted, IAS for Incorrect Attribute Selected.

constraint modules, as shown by the low precision of the "hybrid" modality in the confusion matrices in Figure 5. As a result, we decided to treat the "hybrid" modality as an "uncertain" modality attribute. Hence, the diversity constraints in the "hybrid" setting promote the usage of all three modalities.

## 5 Analysis

**How sensitive is SEER to variations in constraint parameters $\alpha$ and $\beta$?**
We analyze SEER's and $\text{SEER}_{gold}$'s sensitivity to different $(\alpha,\beta)$ value pairs. By increasing $\alpha$, we encourage the selection of exemplars that share the predicted attributes of the test instance. Conversely, increasing $\beta$ promotes the inclusion of exemplars with different attributes, thus ensuring diversity and mitigating errors introduced by the constraint modules. We evaluate the EA and EM on the dev set for the pairs (50,25), (75,0), (75,25), (100,0). The results, depicted in Figure 6, indicate that the difference in EA and EM between the lowest and

|  | FinQA | | TAT-QA | | | |
|  | Modality | | Modality | | Answer type | |
|  | Dev | Test | Dev | Test | Dev | Test |
|---|---|---|---|---|---|---|
| Fine-Tuned$_{BERT}$ | **63.98** | **58.93** | 56.89 | 55.28 | **95.69** | **91.79** |
| ICL$_{CODEX}$ | 59.9 | 58.15 | **63.19** | **62.17** | 87.23 | 87.23 |

Table 2: Accuracy (%) of constraint modules on the dev and test sets.

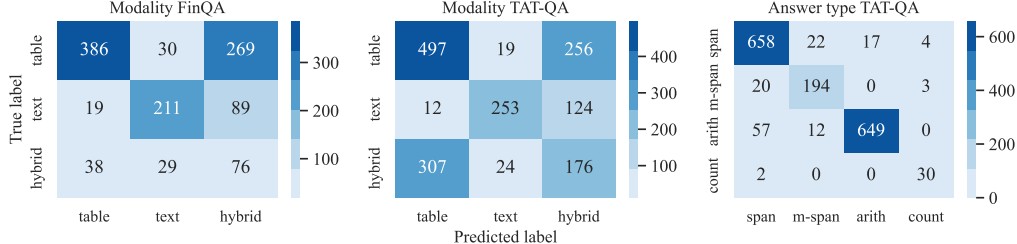

Figure 5: Confusion matrices of the best constraint modules on the test sets.

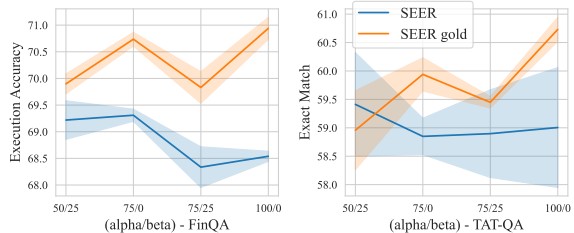

Figure 6: EM and EA (%) for different pairs of ($\alpha/\beta$) values. Results averaged over 5 iterations on the dev set.

highest performing configurations is less than 1%. Consequently, we conclude that SEER does not require extensive tuning of these two parameters to achieve satisfactory performance. SEER$_{gold}$ performs best when $\alpha$ is set to higher values and $\beta$ is set to 0. As SEER$_{gold}$ leverages the ground truth attribute, there is no need to mitigate attribute errors with $\beta$. This finding aligns with our intuition that guiding exemplar selection based on problem attributes improves performance.

**How does SEER perform under different token capacity budgets?**

To evaluate the benefits of the double capacity constraint, we evaluate SEER under different token capacity budgets. While the default token capacity of CODEX is 4096, multiple reasons might lead to a reduction in that value, including using another LLM or financial and time constraints. In the following, we consider capacities of 2048 and 1024 tokens. The 1024 token budget is too

restrictive to even fit most TAT-QA test instances. Hence, we only consider FinQA for that setting. Our experiment involves varying values of M, representing the maximum number of exemplars to be included in the selection. We compare SEER with KATE, the strongest baseline, which disregards token capacity budgets during the search for the most similar exemplars. Figures 7 and 8 report the coverage, i.e. the number of prompts that respect the token budget, and the performance of SEER and KATE. SEER consistently outperforms KATE across all capacity budgets, showcasing a margin of improvement of up to 26% for the more restrictive budgets. While imposing budget restrictions inevitably results in reduced performance, SEER demonstrates a superior ability to mitigate the negative impact compared to KATE. Those results could further be improved with a better estimate of the maximum token capacity $L$. The current estimate is very conservative, as it saves enough space to generate the longest solution of the training sets, 309 and 800 tokens for FinQA and TAT-QA, respectively. Most real-world solutions will require fewer tokens than this upper bound.

**Qualitative Error analysis**

We sample 100 incorrect instances from the test sets at random and categorize them manually in different error categories, summarized in Table 3. On TAT-QA, incorrect values occur even when the exemplars use the required formula. Hence, it seems the error results from the limitations of

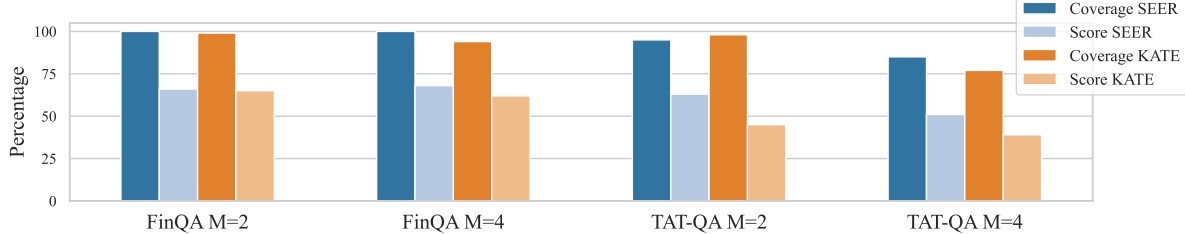

Figure 7: Performance of SEER under a 2048 token capacity budget, as evaluated on the dev set. The coverage is the percentage of exemplar set respecting the budget. The score is the EA (%) for FinQA and EM (%) for TAT-QA.

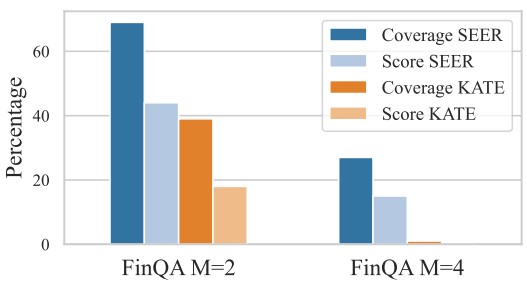

Figure 8: Performance of SEER under a 1024 token capacity budget, as evaluated on the dev set of FinQA.

| Error category | FinQA | TAT-QA |
|---|---|---|
| Values error | 76% | 63 % |
| Operators error | 45% | 30 % |
| Ground truth error | 10 % | 0 % |
| Answer formatting | 0 % | 10 % |
| Execution error | 4 % | 0 % |

Table 3: Percentage of errors per category.

CODEX. This is particularly true for Counting problems, where 4 or more relevant exemplars are provided, but CODEX fails to make the correct count for the test instance. 25% of the FinQA value errors result from a poor computation of the similarity with the candidate exemplars, based on semantically superficial elements. Furthermore, we observed that 17% of the FinQA operator errors are due to one missing reasoning step, e.g. the addition of a constant term. The analysis highlights directions for future work. (1) SEER could benefit from fine-tuning methods to compute the similarity weights (Rubin et al., 2022). (2) SEER can be augmented with mechanisms that predict and control the required number of reasoning steps.

## 6 Conclusion

This paper investigates the problem of exemplar selection for ICL in HybridQA tasks. We propose ILP as a framework for exemplar selection and introduce SEER, a novel method based on Knapsack programs. While existing methods only focus on the high diversity of questions and how to solve them, SEER explicitly tackles two other key challenges of HybridQA exemplar selection in the form of integer linear constraints. Specifically, SEER focuses on the following challenges, the low correlation between the problem's questions and attributes on one hand and the large context required to solve these problems through diversity and capacity constraints. Diversity constraints enhance performance by selecting exemplars that share the same attributes as the test instance. Capacity constraints provide fine-grained control to the end-user over the token budget allocated to the prompt, ensuring sufficient tokens to generate the desired output. This level of control is beneficial in overcoming the limitations of the backbone LLM and dealing with financial constraints. Evaluation on two financial HybridQA datasets shows that SEER outperforms previous ICL exemplar selection methods, especially under limited token capacity.

For future research, we plan to explore additional attributes and seek ways to enhance the overall performance of the constraint modules. Leveraging the flexibility of the Knapsack framework, we intend to study the potential of incorporating constraints defined by end users, such as domain experts, to further refine exemplar selection. Additionally, SEER can be extended beyond the HybridQA setting to encompass other modalities like images and knowledge graphs, as well as other tasks, such as hybrid fact-checking (Aly et al., 2021), data-to-text generation (Parikh et al., 2020), and multimodal fraud detection (Wang et al., 2023).

## Limitations

We identify two main limitations to the method introduced in this paper.

Firstly, we assumed that we could select ICL exemplars from thousands of training instances with annotated textual reasoning. However, in real-world scenarios, obtaining these textual annotations is a time-consuming process that involves manual labeling. In a recent study by Su et al. (2023), a method called *vote-k* was proposed to address this challenge. It involves selecting candidates to annotate from a large unlabeled pool before performing similarity-based exemplar selection. By incorporating *vote-k* into SEER, we can reduce the reliance on a fully annotated training set.

Secondly, while efficient solvers exist for finding the optimal solution to SEER's Knapsack program, it remains an NP-hard problem that can take exponential time to complete in the worst case. To mitigate this issue, we impose a computation budget by setting a timeout of 5 seconds for the solver's execution. We empirically observed that SEER convergence to the global optimum is slower in cases where many candidates have a high similarity weight with the test instance.

## Ethics Statement

**Intended Use** HybridQA demonstrates extensive applicability in real-world scenarios, particularly in the fields of finance and education. In finance, it can reduce the time needed for financial analysts and journalists to process large volumes of financial reports. Additionally, it enhances accessibility to finance-related information for a wider audience. In the education sector, it assists students in solving textbook exercises that involve a hybrid context.

**Misuse potential** While we have not identified any direct potential for misuse of SEER in particular, it is crucial to acknowledge some general concerns with using automatic question answering systems and LLMs. Automatic question answering systems are not infallible. Blindly relying on the outputs generated by the model without double-checking the results can have adverse consequences for end-users, particularly in financial investment analysis. We recommend careful and critical usage of the content generated by LLMs, both with and without SEER. To mitigate this risk, we see potential in research on verifiers for program generation (Ni et al., 2023) and on new explainability methods for LLMs.

## Acknowledgements

This research has been funded by the Statistics Flanders research cooperation agreement on Data Science for Official Statistics and supported by the LOEWE initiative (Hesse, Germany) within the emergenCITY center. We express our gratitude to the Policy Research unit at OpenAI for providing access to the language model CODEX via the OpenAI Research Access Program.

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

## A  The SEER algorithm

SEER is explained in pseudo-code in Algorithm 1.

## B  Diversity constraint templates

We present the potential configurations of diversity constraints, which are determined by the predictions of the constraint modules. There exists one configuration per possible attribute value.

1) Predicted modality: table

$$
\begin{aligned}
\text{subject to} \quad & \sum_{i \in S} table_i \ x_i \geq \alpha \, M \\
& \sum_{i \in S} other\_i \ x_i \geq \beta \, M \\
\text{where} \quad & other\_i = text_i + hybrid_i
\end{aligned}
$$

2) Predicted modality: text

$$
\begin{aligned}
\text{subject to} \quad & \sum_{i \in S} text_i \ x_i \geq \alpha \, M \\
& \sum_{i \in S} other\_i \ x_i \geq \beta \, M \\
\text{where} \quad & other\_i = table_i + hybrid_i
\end{aligned}
$$

3) Predicted modality: hybrid

$$
\begin{aligned}
\text{subject to} \quad & \sum_{i \in S} tab_i \ x_i \geq \beta \, M \\
& \sum_{i \in S} text_i \ x_i \geq \beta \, M \\
& \sum_{i \in S} hybrid_i \ x_i \geq \beta \, M
\end{aligned}
$$

4) Predict answer type: span

$$
\begin{aligned}
\text{subject to} \quad & \sum_{i \in S} span_i \ x_i \geq \alpha \, M \\
& \sum_{i \in S} other_i \ x_i \geq \beta \, M \\
\text{where} \quad & other_i = mspan_i + arith_i + count_i
\end{aligned}
$$

5) Predict answer type: multi-span

$$
\begin{aligned}
\text{subject to} \quad & \sum_{i \in S} mspan_i \ x_i \geq \alpha \, M \\
& \sum_{i \in S} other_i \ x_i \geq \beta \, M \\
\text{where} \quad & other_i = span_i + arith_i + count_i
\end{aligned}
$$

6) Predict answer type: arithmetic

$$
\begin{aligned}
\text{subject to} \quad & \sum_{i \in S} arith_i \ x_i \geq \alpha \, M \\
& \sum_{i \in S} other_i \ x_i \geq \beta \, M \\
\text{where} \quad & other_i = span_i + mspan_i + count_i
\end{aligned}
$$

7) Predict answer type: count

$$
\begin{aligned}
\text{subject to} \quad & \sum_{i \in S} count_i \ x_i \geq \alpha \, M \\
& \sum_{i \in S} other_i \ x_i \geq \beta \, M \\
\text{where} \quad & other_i = span_i + mspan_i + arith_i
\end{aligned}
$$

## C  TAT-QA Prompt examples

Figure 9 illustrate an instance from the TAT-QA dataset. The answer type is multi-span and the modality is text.

**Algorithm 1** Selection of ExEmplars for hybrid Reasoning (SEER)

**Input** Test instance $X_{test}$, training set $S_{train}$, candidate pool size $k$
**Output** Exemplar selection $E_{selection}$

$\quad E_{candidates} \leftarrow KNN(X_{test}, S_{train}, k)$ $\qquad\qquad\qquad\qquad\qquad$ ▷ KNN cosine similarity
$\quad attributes \leftarrow \{\}$
$\quad \textbf{for } module \in constraint\_modules \textbf{ do}$
$\qquad attributes[module.name] \leftarrow module.predict(X_{test})$
$\quad \textbf{end for}$
$\quad knapsak \leftarrow get\_ilp(X_{test}, attributes)$ $\qquad\qquad\qquad\qquad$ ▷ Generate the Knapsack
$\quad E_{selection} \leftarrow solve(knapsack, E_{candidates})$ $\qquad\qquad$ ▷ Solve the Knapsack program

|  | 2019 | 2018 | 2017 |
|---|---|---|---|
| Fixed Price | 1452.4 | 1146.2 | 1036.9 |
| Other | 44.1 | 56.7 | 70.8 |
| Total sales | 1496.5 | 1202.9 | 1107.7 |

On a **fixed-price contract**, we agree to perform the contractual statement[...] On a **cost-plus type contract**, [...]. On a **time-and-material type contract**, we are paid on the basis of direct labor hours [...]

**Question :** what are the contract types?

**Answer :** ['fixed-price type', 'cost-plus type', 'time-and-material type']
**Derivation :** ''
**Scale :** ''

**Python code :**
ans = ['fixed-price type', 'cost-plus type', 'time-and-material type']

Figure 9: TAT-QA instance example

## D  Implementation details

Table 4 presents an overview of the parameters employed in the components of the SEER framework.

## E  Complement on evaluation metrics

The ground truth answers of FinQA and TAT-QA have specific syntactic rules that require access to many training examples to be learned. Those rules are not related to the semantic correctness of the answer. ICL approaches are limited to a few exemplars. Hence, they have a limited ability to learn those syntactic rules. As a result, we provide some flexibility in the evaluation scripts. The following rules are applied: equivalence of answers written as percentage or decimal, removal of characters ($,"million","billion", ...), equivalence up to a rounding of 2 decimals, removal of trailing 0s after the comma. Examples are shown on Table 5

## F  Comparison with Fine-Tuned models

Tables 6 and 7 list the best-reported performance of models that followed a fine-tuning strategy. SEER outperforms several models but lags in performance compared to the current SOTA. There are several trade-offs to consider between fine-tuning and ICL approaches. Fine-tuning requires updating the model weights, which costs time and resources. On the other hand, ICL models can be adjusted to

| Parameter | Value |
|---|---|
| **Preprocessing** | |
| Spacy model | 'en_core_web_lg' |
| NER model | 'bert-base-NER-uncased' |
| **Similarity computation** | |
| Backbone | 'all-mpnet-base-v2' |
| **Fine-tuned constraint modules** | |
| Model | 'bert-base-cased' |
| Epochs | 3 |
| Batch size | 32 |
| Learning rate | $5e^{-5}$ |
| **ICL constraint modules** | |
| Model | 'code-davinci-002' |
| Temperature | 0 |
| Top P | 1 |
| Max tokens | 5 |
| **SEER FinQA** | |
| $k$ | 200 |
| M | 4 |
| Constraint module | ['Modality'] |
| $\alpha$ | 75% |
| $\beta$ | 0 % |
| solver | 'GUROBI CMD' |
| **SEER TAT-QA** | |
| $k$ | 200 |
| M | 8 |
| Constraint module | ['Modality','Answer type'] |
| $\alpha$ | 50% |
| $\beta$ | 25 % |
| solver | 'GUROBI CMD' |
| **Text retrieval** | |
| Backbone | 'all-MiniLM-L6-v2' |
| $k$ | 10 |
| **Code Generation** | |
| Backbone | 'code-davinci-002' |
| Temperature | 0 |
| Top P | 1 |
| Max tokens | 256 |

Table 4: Parameter values

| | |
|---|---|
| Ground Truth | ((4.1 - 3.9) / 3.9) * 100 |
| Prediction | ((4.1 - 3.9) / 3.9) |
| Ground Truth | ['$ 3.4 million'] |
| Prediction | ['3.4 million'] |
| Ground Truth | ['29.0','27.0'] |
| Prediction | ['29','27'] |
| Ground Truth | 45.4 |
| Prediction | 45.3981 |

Table 5: Examples of differences between ground truth and predictions that are counted as correct by the evaluation scripts.

| Model | EA |
|---|---|
| APOLLO$_{ensemble}$ (Sun et al., 2022) | 71.07 |
| ELASTIC (Zhang and Moshfeghi, 2022) | 68.96 |
| **SEER** + CODEX | **68.85** |
| APOLLO(Sun et al., 2022) | 67.99 |
| ReasonFuse (Xia et al., 2023) | 66.17 |
| SoarGraph (Zhu et al., 2023) | 64.5 |
| FinQANet (Chen et al., 2021c) | 61.24 |

Table 6: FinQA Fine-Tuned models vs SEER's performance on the test set (%).

| Model | EM |
|---|---|
| RegHNT (Lei et al., 2022) | 73.6 |
| SoarGraph (Zhu et al., 2023) | 70.3 |
| UniRPG (Zhou et al., 2022b) | 70.2 |
| **SEER** + CODEX | **69.68** |
| GANO (Nararatwong et al., 2022) | 68.4 |
| FinMath (Li et al., 2022) | 60.5 |
| Poet-SQL (Pi et al., 2022) | 59.1 |
| TaCube (Zhou et al., 2022a) | 53.9 |
| TagOp (Baseline) (Zhu et al., 2021) | 55.2 |

Table 7: TAT-QA Fine-Tuned models vs SEER's performance on the dev set (%).

new use cases without retraining. It is interesting to note that both approaches are still far from the human expert performance, which is 91.16% of EA for the test of FinQA and 84.1% of EM for the private test set of TAT-QA. Hence, despite a relatively superior performance of the SOTA fine-tuned models over ICL models, both approaches are equally far from the human performance. Given the recent progress made with LLMs, we believe that the importance of ICL will only increase in the future. Hence, developing strategies for exemplar selection like SEER is an important direction for future work.

# G Significance tests

We evaluate the statistical significance of the results of Table 1 using the Wilcoxon Signed-Rank non-parametric test. The significance is evaluated by comparing the average EA and EM over 3 iterations for FinQA and TAT-QA respectively. Results are reported in Tables 8 and 9.

| | Random | CSP | Fixed set | KATE | Diverse KATE | PromptPG | SEER |
|---|---|---|---|---|---|---|---|
| CSP | 1e-15 | - | - | - | - | - | - |
| Fixed set | 2e-27 | 2e-3 | - | - | - | - | - |
| KATE | 2e-29 | 6e-6 | 0.02 | - | - | - | - |
| Diverse KATE | 4e-32 | 1e-5 | 0.03 | 0.68 | - | - | - |
| PromptPG | 4e-13 | 8e-16 | 4e-24 | 3e-29 | 1e-29 | - | - |
| SEER | 3e-35 | 3e-9 | 1e-4 | 7e-3 | 2e-3 | 2e-36 | - |
| SEER$_{gold}$ | 3e-38 | 6e-10 | 5e-5 | 2e-3 | 6e-3 | 6e-37 | 0.24 |

Table 8: Significance tests for FinQA

| | Random | CSP | Fixed set | KATE | Diverse KATE | PromptPG | SEER |
|---|---|---|---|---|---|---|---|
| CSP | 1e-13 | - | - | - | - | - | - |
| Fixed set | 5e-41 | 1e-20 | - | - | - | - | - |
| KATE | 2e-49 | 1e-20 | 0.012 | - | - | - | - |
| Diverse KATE | 1e-20 | 1e-3 | 1e-4 | 8e-14 | - | - | - |
| PromptPG | 3e-9 | 2e-5 | 9e-30 | 6e-36 | 8e-12 | - | - |
| SEER | 2e-51 | 4e-23 | 2e-3 | 0.24 | 1e-15 | 4e-40 | - |
| SEER$_{gold}$ | 4e-56 | 2e-28 | 3e-6 | 4e-3 | 1e-19 | 5e-45 | 0.015 |

Table 9: Significance tests for TAT-QA