# OpenReview forum: "SEER : A Knapsack approach to Exemplar Selection for In-Context HybridQA"
_EMNLP/2023/Conference — EMNLP 2023 Main_

### Official Review · Reviewer_Fa3v · 2023-08-04

**Soundness:** 4

**Excitement:**

4: Strong: This paper deepens the understanding of some phenomenon or lowers the barriers to an existing research direction.

**Paper Topic And Main Contributions:**

This paper addresses a key issue in hybridQA (text+table): exemplar selection. It treats this as a knapsack problem, which allows the system to more effectively select exemplars under various resource constraints (primarily length of input to an LLM).
The paper provides an excellent introduction to the task and to ICL and Knapsack approaches. The results are good and are compared to multiple other systems on two standard datasets.


**Questions For The Authors:**

end of section 3.2: Include the best fine-tuning approach in the comparison here; that is mention it in the text (there will be room once you do not refer to the appendix) and add a row to the main results table

lines 399-402: at least state whether the non-gold SEER results are stat sig better than the next best system.

Figure 4:  some of the results look very similar: are they stat sig different? Also, indicate the variance in the 3 runs (e.g. have a min/max bar overlaying the current ones).

I think the knapsack approach could be used for other tasks that use ICL with LLM. Perhaps mention this in the conclusion as an area for future work/future impact of the proposed method.


**Reasons To Accept:**

- Excellent use of knapsack algorithm to the problem.
- Ability to adjust to input length constraints on LLMs.
- The two above abilities are likely to be applicable to other NLP problems.
- Solid results and comparison to strong SOTA competitors.
- Well written and easy to understand.

**Reasons To Reject:**

- Need to include a few items from the appendix into the main paper (minor and I think these can fit).
- No qualitative error analysis which could help in determining how to augment the approach in the future.
- Huge appendix, which is not realistic to ask reviewers to look at.

**Reproducibility:**

3: Could reproduce the results with some difficulty. The settings of parameters are underspecified or subjectively determined; the training/evaluation data are not widely available.

**Reviewer Confidence:**

4: Quite sure. I tried to check the important points carefully. It's unlikely, though conceivable, that I missed something that should affect my ratings.

**Typos Grammar Style And Presentation Improvements:**

The paper is very well laid out and easy to follow.
The English is generally excellent but there are a few systematic errors:
- do not have a space before colons (:) in English
- modifiers in noun-noun modification should be in the singular even if they refer to a plural (e.g. "candidate set" not "candidates set")
- "allow to + infinitive" requires an object (line 252: allows the user to balance)
- "train set" and "train instance" => "training set" and "training instance"

line 059: problem( => problem (
line 090: insert space after commas
line 225: what does "neighbor of the test instance's" mean? I think this is just a typo; please rephrase
line 320: syntactical => syntactic
line 375: estimates of => estimate of
line 379: solver3to => solver3 to

Table 1: It will be easier to read if the rows are ordered roughly worst to best system.

Section 5: This could be earlier. Some of the explanations are helpful in understanding ICL and ILP if the reader is not familiar with them.

line 620: potentialWhile => potential While

Ethical concerns: I am glad the authors are thinking about possible misuse. What is cited is pretty much a standard concern. It is good to cite it, but perhaps mention more strongly that this is a general concern with using LLM and also for using automatic QA without double-checking the results.

---

> ### Author Rebuttal · Authors · 2023-08-28
>
> Dear reviewer, thank you for your constructive comments and detailed improvements suggestions. We are encouraged that you see in SEER an **excellent application of the Knapsack algorithm** to exemplar selection. We thank you for pointing out the **solid results** and **comparison with strong competitors**, also supported by Reviewer dh6X.
>
> We address your remarks and questions below:
>
> > **Need to include a few items from the appendix into the main paper (minor and I think these can fit). Huge appendix, which is not realistic to ask reviewers to look at.**
>
> We acknowledge that the current appendix is too long and some content needs to move to the main text. We **will take the following actions to correct this** :
> - The sections dealing with preprocessing (App. G), the description of PromptPG (App. J), the confusion matrices and description of the constraint modules (App. K and L), and the error analysis (App. O) are moved to the main body.
> - Certain content, such as dataset statistics and the instance examples from the FinQA and TAT-QA datasets (App. B,C,F), is already well-documented in the original dataset papers. Hence, we remove them from the Appendices.
> - We will add the best fine-tuning approach (App. M) and the significance scores of SEER and KATE (App. N) directly into the main body.
>
> By making those modifications, the Appendices are reduced to 3 pages, and will contain only the optional content in the camera-ready version of the paper.
>
> > **No qualitative error analysis which could help in determining how to augment the approach in the future.**
>
> Thank you for suggesting the **addition of a qualitative error analysis**. We had previously sampled 100 error cases per dataset and classified the errors in different categories (Table 11). Based on your recommendation, we explored again the sampled error cases together with their exemplar set and obtained **three major insights**:
> - 25% of the FinQA errors resulted from a poor ranking of the candidate exemplars based on their similarity with the test instance. SEER could benefit from more advanced methods to compute the similarity weights in the Knapsack objective function. Those are currently computed with a pre-trained Sentence-BERT model, but **more advanced fine-tuning** methods are available (Rubin et al., 2022).
> - 17% of the FinQA errors had one missing reasoning step, e.g. the addition of a constant term. In the future, the approach can be augmented with mechanisms that predict and control the **required number of reasoning steps**.
> - When a TAT-QA error contains incorrect values in the reasoning (31%), the exemplars often possess the correct attributes and have a high similarity with the test instance. Hence, it seems the error results from the limitations of the LLM itself, i.e., CODEX in our case. This is particularly true for Counting problems, where 4 or more relevant Counting exemplars are provided, but CODEX still fails to make the correct count for the test instance.
>
> We will **include the qualitative error analysis** and the insights **in the main body** of the revised paper.
>
> > **The two above abilities are likely to be applicable to other NLP problems. I think the knapsack approach could be used for other tasks that use ICL with LLM. Perhaps mention this in the conclusion as an area for future work/future impact of the proposed method.**
>
> We are heartened by your recognition of the **broader applicability of SEER's knapsack approach** in NLP applications and beyond. The current conclusion already highlights Hybrid Fact-Checking (Aly et al., 2021) as an avenue for future work. We will augment the conclusion to encompass other promising applications, such as Data-to-Text Generation (Parikh et al., 2020) and Multimodal Fraud Detection (Wang et al., 2023).
>
>
> >  **Include the best fine-tuning approach in the comparison here; that is mention it in the text (there will be room once you do not refer to the appendix) and add a row to the main results table. lines 399-402: at least state whether the non-gold SEER results are stat sig better than the next best system**
>
> Thank you for this recommendation. As mentioned above, we will add the best fine-tuning approach (App. M) and the significance scores of SEER and KATE (App. N) directly into the main body, in the main results section.
>
> >**Figure 4: some of the results look very similar: are they stat sig different? Also, indicate the variance in the 3 runs (e.g. have a min/max bar overlaying the current ones).**
>
> Thank you for suggesting those additions that reinforce the insights observed from the subset analysis. We evaluate the statistical significance with the Wilcoxon Signed-rank non-parametric test over the 3 runs.  SEER and KATE are **stat sig different (p-value=0.05)** for Correct Attribute Predicted (CAP) and Correct Attribute Selected (CAS). SEER and the Fixed set baseline are stat sig different for CAP, CAS, and Incorrect Attributed Predicted (IAP). KATE and Fixed set are stat sig different for CAP, CAS, and IAP. The variance over the 3 runs is low, except for the KATE baseline on CAS and IAP, and for the Fixed set baseline on IAS. The mean and standard deviation values are reported in the following table.
>
> |    | CAP |  CAS |  IAP |  IAS  |
> | ----- | ----  | ----  | ----   | ----   |
> |**SEER** | 75.48+-0.82 | 71.83+-0.94 | 59.42+-0.47 | 48.53+-0.47|
> | **KATE** | 72.8 +-0.47 |  69.77+-0.94 | 58.86+-0.0 | 48.75+-1.25 |
> | **Fixed set** | 70.03+-0.94 | 66.1+-2.16 | 55.56+-1.89 | 50.11+-0.47 |
>
>
> We will edit Figure 4 to **include min/max bars for the standard deviations**.
>
>
> > **The paper is very well laid out and easy to follow. The English is generally excellent but there are a few systematic errors.**
>
> Thank you for taking the time and effort to identify errors and **providing pointers** to their position in the text, which will **greatly assist us** in rectifying them. We will edit all typos and syntactic mistakes for the camera-ready version.
>
> >**Related work could be earlier. Some of the explanations are helpful in understanding ICL and ILP if the reader is not familiar with them.**
>
> Thank you for this suggestion. We will move the related work section between Section 1 Introduction and Section 2 Methodology.
>
> >**I am glad the authors are thinking about possible misuse. What is cited is pretty much a standard concern. It is good to cite it, but perhaps mention more strongly that this is a general concern with using LLM and also for using automatic QA without double-checking the results.**
>
> We propose the following modification to the original text : “While we have not identified any direct potential for misuse for SEER in particular, it is crucial to acknowledge **some general concerns** with using automatic question answering systems and LLMs. Automatic question answering systems are not infallible. Blindly relying on the outputs generated by the model without double-checking the results can have adverse consequences for end-users, particularly in financial investment analysis. [...]”
>
>
> References
>
> - Rami Aly, Zhijiang Guo, Michael Schlichtkrull, James Thorne, Andreas Vlachos, Christos Christodoulopoulos, Oana Cocarascu, and Arpit Mittal. 2021. Feverous: Fact extraction and verification over unstructured and structured information. In Proceedings of the Neural Information Processing Systems Track on Datasets and Benchmarks, volume 1. Curran.
> - Ankur Parikh, Xuezhi Wang, Sebastian Gehrmann, Manaal Faruqui, Bhuwan Dhingra, Diyi Yang, and Dipanjan Das. 2020. ToTTo: A Controlled Table-To-Text Generation Dataset. In Proceedings of the 2020 Conference on Empirical Methods in Natural Language Processing (EMNLP), pages 1173–1186, Online. Association for Computational Linguistics.
> - Ohad Rubin, Jonathan Herzig, and Jonathan Berant. 2022. Learning To Retrieve Prompts for In-Context Learning. In Proceedings of the 2022 Conference of the North American Chapter of the Association for Computational Linguistics: Human Language Technologies, pages 2655–2671, Seattle, United States. Association for Computational Linguistics.
> - Gang Wang, Jingling Ma, and Gang Chen. 2023. Attentive statement fraud detection: Distinguishing multimodal financial data with fine-grained attention. Decision Support Systems, 167:113913

---

### Official Review · Reviewer_aS63 · 2023-08-05

**Soundness:** 4

**Excitement:**

3: Ambivalent: It has merits (e.g., it reports state-of-the-art results, the idea is nice), but there are key weaknesses (e.g., it describes incremental work), and it can significantly benefit from another round of revision. However, I won't object to accepting it if my co-reviewers champion it.

**Paper Topic And Main Contributions:**

The paper proposes Knapsack Programs as a framework to model exemplar selection for ICL. For a given test instance, a Knapsack Program is solved to obtain the optimal exemplars set. This expressive framework allows balancing the diversity and similarity of the selected exemplars while controlling the prompt size with user-defined linear constraints. The paper introduces SEER, a novel method to select exemplars for HybridQA using Knapsack Programs. SEER reduces the candidates set with a nearest neighbor filtering, and leverages constraint modules to predict the attributes of the test instance. The experimental results show that the proposed method outperforms previous exemplar selection methods.

**Reasons To Accept:**

1. The code is provided.
2. clear motivation.

**Reasons To Reject:**

1. The paper, especially the figures, is not formed in a good layout.
2. Analysis experiments in the main body is not adequate.\
3. The writing and story is not easy-reading.

**Reproducibility:**

4: Could mostly reproduce the results, but there may be some variation because of sample variance or minor variations in their interpretation of the protocol or method.

**Reviewer Confidence:**

4: Quite sure. I tried to check the important points carefully. It's unlikely, though conceivable, that I missed something that should affect my ratings.

---

> ### Author Rebuttal · Authors · 2023-08-28
>
> Dear reviewer, thank you for your helpful comments. We appreciate that you see a **clear motivation** in our proposed approach. SEER addresses a **key issue** in HybridQA (Reviewer Fa3v). We are the first to show that the Knapsack is well-suited for HybridQA problems that combine text, table, and hybrid data (Reviewer Fa3v, Reviewer dh6X).
>
> We address your remarks and questions below:
>
> > **The paper, especially the figures, is not formed in a good layout.**
>
> We acknowledge that certain figures might not have achieved the desired level of clarity in their current presentation. This issue **can be resolved** for the camera-ready version of the paper. We intend to reduce the amount of text shown on the illustrative examples of FinQA and TAT-QA (Figure 1, 2, 10) to make them more visually-engaging. We **will apply those changes** to the camera-ready version of the paper.
>
> > **Analysis experiments in the main body is not adequate.**
>
> The analysis experiments in the main body consist of two parts. Allow us to elaborate on the **significance and rationale** behind these analyses, addressing their relevance and contributions.
> - The first analysis is concerned about controlling the length of the prompt while adhering to **token capacity constraints**. This aspect is vital due to the **inherent limitations of LLMs** in processing a finite number of tokens as input. Moreover, commercial LLMs, such as GPT-4, often use the token count as a pricing unit, rendering **token budgeting** a practical consideration for organizations. The **importance of this property**, and its applicability to other NLP tasks, is supported by  Reviewer Fa3v. In our detailed analysis, we compare SEER with its closest baseline, KATE, under different token budgets. We show how SEER's strategic incorporation of token capacity constraints yields **superior performance** compared to KATE, particularly in scenarios with limited token capacity budgets.
> - The second analysis evaluates SEER’s **sensitivity** to the values of the **hyperparameters** $\alpha$ and $\beta$. Those parameters govern the selection of exemplars based on their shared attributes with the test instance. Our analysis shows that the sensitivity to changes in hyperparameter values is low, meaning that SEER can produce **good results without extensive hyperparameter tuning**.
>
> In essence, both analyses **contribute substantially** to SEER's assessment, addressing robustness and applicability.
>
> Following the recommendation of Reviewer Fa3v’s, we will also include a **qualitative error analysis** in the revised version of the paper. This analysis will highlight research directions to augment the approach in future work.
>
> > **The writing and story is not easy-reading**
>
> Thank you for your feedback on the readability of the paper. We fully recognize the importance of ensuring a smooth and comprehensible reading experience for our audience. Could we ask you to provide pointers to the parts of the paper that are currently not easy-reading? We are **committed to addressing these concerns** for the camera-ready version.
>
> We will take the following actions to improve the presentation :
> - **Reordering of Sections**: We will relocate the "Related Work" section after "Introduction" and before "Methodology," in line with the feedback provided by Reviewer Fa3v. This adjustment will provide readers with background information on In-Context Learning and Integer Linear Programming, improving their understanding of the subsequent content.
> - **Visual clarity**: We will reduce the amount of text in the figures that illustrate FinQA and TAT-QA instances (Figures 1, 2, 10) . This modification aims to render the visual representation more engaging for readers.
> - **Results Presentation**: In response to the suggestion by Reviewer Fa3v, we will arrange the baselines in the results table from the worst to the best performing. This arrangement will provide a more intuitive representation of the relative performance of the different methods.

---

### Official Review · Reviewer_dH6X · 2023-08-07

**Typos Grammar Style And Presentation Improvements:** 1. The "correct answer" for Figure 19…
**Soundness:** 4

**Excitement:**

4: Strong: This paper deepens the understanding of some phenomenon or lowers the barriers to an existing research direction.

**Paper Topic And Main Contributions:**

The paper proposes an approach for selecting examples for few-shot prompting. They propose defining an optimization problem using Knapsack based ILP. The objective is to maximize the similarity of the example with the test example subject to variable number of constraints. The constraints include budget of maximum number of tokens, maximum number of allowed examples. They also include constraints on diversity of selected examples. One of the constraints is on the modality of the example, the modality being one of table, text, or both. Another constraint is on the type of answer including span, multi-span, and so on.
They evaluate on the task of financial QA including FinQA and TAT-QA

The contributions include:
1. They show that proposing such a knapsack based ILP optimization outperforms other baselines including random selection of examples, k-nearest neighbor search.
2. They show that the approach is effective for dataset which combines different forms of input including text, table, or both.

**Questions For The Authors:**

1. Why are the results for PromptPG so low?
2. A baseline could be to divide the M number of examples into two groups, one of each modality, followed by kNN search for each group. I think this baseline could be stronger than KATE, avoids retrieving no examples of the correct modality as pointed in Figure 20, 21. What do you think?


**Reasons To Accept:**

1. Stronger results compared to baselines

**Reasons To Reject:**

1. The paper has a huge appendix which is not entirely optional.

**Reproducibility:**

3: Could reproduce the results with some difficulty. The settings of parameters are underspecified or subjectively determined; the training/evaluation data are not widely available.

**Reviewer Confidence:**

4: Quite sure. I tried to check the important points carefully. It's unlikely, though conceivable, that I missed something that should affect my ratings.

---

> ### Author Rebuttal · Authors · 2023-08-28
>
> Dear Reviewer, thank you for your constructive and insightful feedback. We are glad that you recognize the contribution of SEER to solving HybridQA problems and the improvements it brings over existing baselines.
> We address your remarks and questions below:
>
> > **The paper has a huge appendix which is not entirely optional.**
>
> We acknowledge that the current appendix is too long and contains information that should be moved to the main text. However, as mentioned by Reviewer Fa3v, this can be solved with minor changes. The revision will contain the **following modifications** :
> - The sections dealing with preprocessing (App. G), the description of PromptPG (App. J), the confusion matrices and description of the constraint modules (App. K and L), and the error analysis (App. O) are moved to the main body.
> - Certain content, such as dataset statistics and the instance examples from the FinQA and TAT-QA datasets (App. B, C, F), is already well-documented in the original dataset papers. Hence, we remove them from the Appendices.
> - We will add the best fine-tuning approach (App. M) and the significance scores of SEER and KATE (App. N) into the main body.
>
> By making those modifications, the Appendices are reduced to 3 pages, and will contain only the optional content in the camera-ready version of the paper.
>
> > **Why are the results of PromptPG so low?**
>
> PromptPG (Lu et al., 2023) trains a neural network with the REINFORCE (Williams, 1992) policy gradient algorithm to select exemplars from a fixed candidate pool. Like all reinforcement learning algorithms, it needs to balance **exploration** and **exploitation** to move towards a global optimum. Our empirical observations have led us to conclude that PromptPG **under-explores** the candidate pool. It tends to prematurely settle into a fixed suboptimal selection of exemplars after a limited number of learning episodes. Given
> that the initial candidate pool is randomly selected, the performance of a fixed subset is close to the random baseline. From those observations, we conclude that PromptPG needs adjustment beyond the original configuration proposed in Lu et al. (2023).
>
>
> >**A baseline could be to divide the M number of examples into two groups, one of each modality, followed by kNN search for each group. I think this baseline could be stronger than KATE, avoids retrieving no examples of the correct modality as pointed in Figure 20, 21. What do you think?**
>
> Your insightful suggestion raises an interesting new baseline approach that has yet to be explored in our study! Intriguingly, we find that only 42% of the exemplar set selected by KATE on FinQA incorporates all modalities. Consequently, there is a **clear potential** for this new baseline, which we call **Diverse KATE**, to outperform KATE.
> It is worth mentioning that this baseline closely resembles SEER when the hyperparameters $\alpha$ and $\beta$ are both set to 0.5. In that scenario, SEER requires to select an equal number of exemplars possessing the predicted attribute and those with a different attribute.
> We have conducted an initial evaluation of a single iteration of the Diverse KATE approach on the FinQA dataset. We obtained an **Execution Accuracy** (EA) score of 67.12,  a small improvement over the EA of KATE (67.07). We will conduct additional rounds of evaluation for both the FinQA and TAT-QA datasets. We will **include and discuss the obtained results** in the camera-ready version of our paper.
>
> > **The "correct answer" for Figure 19 should be 200+500+750=1450 instead of 1000?**
>
> Thank you for identifying this typo. The correct answer, also predicted by SEER, is indeed 1450. This corresponds to the instance with the identifier “ETR/2008/page_298.pdf-2” from the FinQA test set.
>
> > **It is not clear from the description in the paper about how the tables are encoded? Is there any specific format for linearizing the table?**
>
> Thank you for pointing this out. The tables are linearized with **“|” as column delimiters** and **“\n” as row delimiters**. This is a standard scheme for table encoding (Lu et al., 2023), and it offers the direct advantage of introducing minimal additional tokens.
> To provide greater clarity, we will explain the table encoding in Section 4.1 Datasets.
>
> > **In line 418, should it be CAS \subset CAP?**
>
> Thank you for your keen observation. However, **our notation is accurate as presented**: CAP $\subseteq$ CAS. When the constraint module correctly predicts the attribute of a test instance, it guarantees that said instance will be incorporated into the selection, as stipulated by the $\alpha$ diversity constraint. Consequently, every element within the Correct Attribute Predicted (CAP) set belongs to the Correct Attribute Selected (CAS) set.
> Through the $\beta$ diversity constraint, it is possible that one of the selected exemplars possesses the correct attribute, even if an incorrect attribute was predicted by the constraint modules. As a result, the CAS set also includes elements of the Incorrect Attribute Predicted (IAP) set. Consequently, CAP is a subset of CAS.
>
>
> References
>
> - Pan Lu,Liang Qiu,Kai-Wei Chang,Ying Nian Wu, Song-Chun Zhu,Tanmay Rajpurohit,Peter Clark, and Ashwin Kalyan. 2023. Dynamic Prompt Learning via policy gradient for semi-structured mathematical reasoning. International Conference Learning Representations (ICLR).
>
> - Ronald J Williams. 1992. Simple statistical gradient-following algorithms for connectionist reinforcement learning. Machine Learning,8(3):229–256.

---

### Meta-Review · Area_Chair_Rcoh · 2023-09-18

**Recommendation:** 5

**Metareview:**

This paper presents an ILP formulation for selecting exemplars -- for prompting LLMs in a task of hybrid (text+tables) QA. The idea is interesting and novel. Moreover, it allows for a better control of selected exemplars in terms of their diversity, length/number or other constraints. The code will be publicly available. All the reviewers agree that this contribution is sound and exciting.

However, there are some presentation issues. The main issue is the appendix (reviewers #1 and #3): it is very large and contains essential information. It should be reorganized to make the paper self-sufficient and easy to follow. Minor presentation issues (easily fixable) have been noted by the reviewers.

---

### Decision · Program_Chairs · 2023-10-07

**Decision:**

Accept-Main

**Comment:**

This paper presents an ILP formulation for selecting exemplars -- for prompting LLMs in a task of hybrid (text+tables) QA. The idea is interesting and novel. Moreover, it allows for a better control of selected exemplars in terms of their diversity, length/number or other constraints. The code will be publicly available. All the reviewers agree that this contribution is sound and exciting.

However, there are some presentation issues. The main issue is the appendix (reviewers #1 and #3): it is very large and contains essential information. It should be reorganized to make the paper self-sufficient and easy to follow. Minor presentation issues (easily fixable) have been noted by the reviewers.